# Proteomics as a Metrological Tool to Evaluate Genome Annotation Accuracy Following De Novo Genome Assembly: A Case Study Using the Atlantic Bottlenose Dolphin (*Tursiops truncatus*)

**DOI:** 10.3390/genes14091696

**Published:** 2023-08-25

**Authors:** Benjamin A. Neely, Debra L. Ellisor, W. Clay Davis

**Affiliations:** National Institute of Standards and Technology, NIST Charleston, 331 Fort Johnson Road, Charleston, SC 29412, USA; debra.ellisor@nist.gov (D.L.E.); clay.davis@nist.gov (W.C.D.)

**Keywords:** de novo genome, genome accuracy, proteomics, marine mammal

## Abstract

The last decade has witnessed dramatic improvements in whole-genome sequencing capabilities coupled to drastically decreased costs, leading to an inundation of high-quality de novo genomes. For this reason, the continued development of genome quality metrics is imperative. Using the 2016 Atlantic bottlenose dolphin NCBI RefSeq annotation and mass spectrometry-based proteomic analysis of six tissues, we confirmed 10,402 proteins from 4711 protein groups, constituting nearly one-third of the possible predicted proteins. Since the identification of larger proteins with more identified peptides implies reduced database fragmentation and improved gene annotation accuracy, we propose the metric NP_10_, which attempts to capture this quality improvement. The NP_10_ metric is calculated by first stratifying proteomic results by identifying the top decile (or 10th 10-quantile) of identified proteins based on the number of peptides per protein and then returns the median molecular weight of the resulting proteins. When using the 2016 versus 2012 *Tursiops truncatus* genome annotation to search this proteomic data set, there was a 21% improvement in NP_10_. This metric was further demonstrated by using a publicly available proteomic data set to compare human genome annotations from 2004, 2013 and 2016, which showed a 33% improvement in NP_10_. These results demonstrate that proteomics may be a useful metrological tool to benchmark genome accuracy, though there is a need for reference proteomic datasets across species to facilitate the evaluation of new de novo and existing genome.

## 1. Introduction

Since 2007, there has been a rapid decrease in whole-genome sequencing costs coupled with improved read lengths and developments of long-range techniques such as synthetic long-reads and mapping protocols. Specifically, sequencing costs have plummeted from roughly USD 500 per Mb in 2007 to less than USD 0.01 per Mb, as of 2022 [1]. Concurrently, the access to high performance computing environments has improved along with an endless supply of new genome assembly and annotation tools. With these new resources, it is now possible to rapidly generate an analysis [2]. Excellent early examples of this are two completed mammalian genomes from the mid-2010s (high-quality de novo genomes for non-model organisms to enable the domestic goat, *Capra hircus* [3,4], and the Hawaiian monk seal, *Neomonachus schauinslandi* [5], to be proteomic) that utilized a combination of approaches including optical mapping, synthetic long reads, long read technology and chromatin interaction mapping to generate highly contiguous (scaffold N50 > 29.5 Mbp) de novo genomes at a relatively low cost. This is exemplified by the DNA Zoo and their so-called USD 1000 genome [6]. Overall, the result of these parallel advancements are numerous large-scale sequencing projects [7], with the most ambitious targeting approximately 9000 eukaryotic species (Earth BioGenome Project). With the current inundation of new high-quality de novo genomes, there is a continued need for improved metrics to evaluate genome accuracy.

Genome assemblies and annotations are evaluated in terms of contiguity and completeness, which are both indicators of genome accuracy. Measures of contiguity, such as scaffold N50 or N90 length, typically correspond to the quality of the genome assembly [8]. Scaffold N50 or N90 length is similar to a median or quantile scaffold length but is dependent on assembly size. Greater scaffold contiguity tends to result in more protein-coding sequences and isoforms. For example, one of the initial finished human genome assemblies from 2004 (NCBI Build 34) had a scaffold N50 of 27.2 Mbp and 27,180 protein-coding sequences, which was improved in March 2016 to a scaffold N50 of 59.4 Mbp and 109,018 protein-coding sequences (NCBI Release 108). Gains can be even more pronounced in non-model organisms with improved de novo genome assemblies. For example, the *Alligator mississippiensis* (American alligator) genome improved from a scaffold N50 of 508 kbp to 10 Mbp using updated sequencing methods [9]. Similarly, the focus of this study, *T. truncatus* (Atlantic bottlenose dolphin), improved from a scaffold N50 of 116 kbp to 26.6 Mbp. Studies have shown that assembly contiguity often corresponds to assembly quality [8] but does not necessarily correlate with genome completeness and therefore accuracy [10]. One way to evaluate genome completeness is by using predicted conserved gene products. First used in the Core Eukaryotic Genes Mapping Approach (CEGMA) [11,12], this concept has developed into Benchmarking Universal Single-Copy Orthologs (BUSCO), which is a content-based quality assessment that uses universal single-copy markers to gauge genome completeness [10,13,14,15]. Specifically, BUSCO evaluates genome completeness by comparing genes to a set of orthologous genes of a given clade; thus; humans could be evaluated against orthologous genes from primates and *T. truncatus* against orthologous genes from cetartiodactyla [16]. The resulting score is based on the length of the matches yielding the number of complete, fragmented, or missing genes, thereby giving a measure of “completeness” beyond size and contiguity. It is evident that using many metrics to benchmark de novo genomes is essential to evaluating genome quality. Given the orthogonal nature of proteomics and its dependence on accurately predicted gene annotations, a quality metric based in this analytical domain may be advantageous.

Data-dependent acquisition bottom-up proteomics is one method to confirm gene annotations by observing the predicted proteins using mass spectrometry. First, proteins are digested with a known protease, and the resulting peptides are fragmented within a mass spectrometer. Next, using an accurate mass of the precursor and the resulting fragmentation pattern, search algorithms can probabilistically identify peptides and then infer proteins in the search database. Alternatively, spectral libraries directly match fragmentation patterns, though these initial assignments are typically made using database-dependent approaches [17,18,19]. With the current generation of mass spectrometers, which have high duty cycles with high mass accuracy and resolution, we may be approaching the era of being able to infer the majority of proteins in a genome. For example, even in 2017, a multi-fractionated proteomic analysis of HeLa tissue accounted for 91.5% of gene products measured in the same tissue by RNA-seq (12,209 protein coding sequences versus 13,347 gene products) [20], and more recent proteomic studies have pushed this depth even further with fractionation and multiple proteases, yielding a high coverage of 17,717 protein groups from six cell lines, which are nearly 90% of all possible proteins [21]. Since bottom-up proteomics relies completely on a database for peptide identifications and protein inference, it may be possible that a high-quality mass spectrometric dataset could be used to benchmark genome assembly and annotation quality. Though not the same as the NP_10_ concept described herein, an excellent attempt was made whereby proteomic de novo results are compared to different genome annotations to evaluate protein database suitability [22], which is likewise related to annotation quality. This method is especially useful when choosing between many different protein databases.

The purpose of the current study was two-fold: (i) provide detailed proteomic profiling of a marine mammal and (ii) use these data to evaluate the 2016 *T. truncatus* assembly and annotation. The *T. truncatus* genome assembly was updated by the National Institute of Standards and Technology (NIST) in 2016 to demonstrate improved biomolecular measurement capabilities in *T. truncatus*, such as proteomics, and therefore we have attempted to quantify the improvements. On average, over 4800 proteins (at <1% FDR) were identified in six different tissues, and when combined yielded 10,402 protein identifications. Although not an exhaustive proteomic dataset, this confirmed approximately 1/3 of the predicted protein-coding genes. This dataset is an invaluable resource to support comparative proteomics in diving mammals related to comparative evolution [23] and biomimicry [24] and demonstrates the feasibility of accelerating cutting-edge bioanalytical approaches in non-model organisms (as reviewed by Heck and Neely [2]). Secondly, the new de novo assembly resulted in increased protein identifications but also fewer peptide identifications, despite more than a 200-fold improvement in scaffold N50 over the previous 2012 assembly. We investigated these differences at the peptide and protein level to identify global trends and proposed a new measure of genome annotation quality, NP_10_. This new measure was further demonstrated by evaluating human genome improvements over the decade from 2004 to 2016 using publicly available proteomic data. Although the *T. truncatus* genome has since been updated in 2020 using a genome assembly from the Vertebrate Genome Project and a parallel update in the NCBI RefSeq genome annotation (release 102), we still feel these metrics are relevant for any current and future evaluations of any species. Overall, these results highlight the improved annotation accuracy of the 2016 *T. truncatus* genome and the utility of proteomics as a metrological tool for evaluating genome annotation quality, as well as emphasizes the need for reference proteomic datasets to facilitate metrology in new and existing genomes.

## 2. Materials and Methods

### 2.1. Sample Source and Preparation

Bottlenose dolphin tissues were collected from animals under appropriate permits (Appendix A) and stored at liquid nitrogen temperatures (−150 °C to −180 °C) until cryohomogenization in the National Institute of Science and Technology (NIST) Biorepository [25]. From the resulting fine powder, 5 mg was subsampled and the proteins were extracted using RapiGest (Waters, Milford, MA, USA). Briefly, 150 µL of 0.1% (*w*/*v*) RapiGest (in 50 mM ammonium bicarbonate) was added, resulting in a solution of 33 µg/µL tissue. The extraction mixture was shaken at 600 rpm for 25 min at room temperature followed by a removal of large debris using a benchtop microcentrifuge. From this solution, a 5 µL aliquot was removed and suspended in 35 µL of 0.1% (*w*/*v*) RapiGest (in 50 mM ammonium bicarbonate), followed by the addition of 40 uL of 50 mM ammonium bicarbonate. Next, the sample was reduced with 10 µL of 45 mM dithiothreitol (Sigma, Burlington, MA, USA) (DTT; final concentration of 5 mM) and incubated at 60 °C for 30 min, then allowed to cool to room temperature. The mixture was alkylated using 3.75 µL of 375 mM iodoacetamide (Pierce, Thermo Scientific, Waltham, MA, USA; final concentration of 15 mM) and incubated in the dark at room temperature for 20 min. Prior to the addition of trypsin(Promega, Madison, USA), 100 µL of 50 mM ammonium bicarbonate was added. A 3.3 µL aliquot of trypsin (MS-Grade; 1 µg/µL in 50 mM acetic acid) was added (1:50 trypsin:protein) and samples were incubated overnight at 37 °C. The digestion was halted and RapiGest cleaved with the addition of 100 µL of 3% (*v*/*v*) trifluoroacetic acid (1% final concentration) and incubated at 37 °C for 30 min before centrifugation and removal of the supernatant. Samples were processed using Pierce C18 spin columns (8 mg of C18 resin; Thermo Scientific) according to the manufacturer’s instructions. Each sample was processed in duplicate yielding at maximum of 60 µg peptides. These solutions were evaporated to dryness in a vacufuge then reconstituted in 150 µL of 5% acetonitrile in water.

### 2.2. Mass Spectrometry

Samples were analyzed using an UltiMate 3000 Nano LC coupled to a Fusion Lumos mass spectrometer (Thermo Fisher Scientific, Waltham, MA, USA). Resulting peptide mixtures (10 µL) were loaded onto a PepMap 100 C18 trap column (75 µm id × 2 cm length; Thermo Fisher Scientific) at 3 µL/min for 10 min with 2% (*v*/*v*) acetonitrile and 0.05% (*v*/*v*) trifluoroacetic acid (Pierce, Rockford, USA) followed by separation on an Acclaim PepMap RSLC 2 µm C18 column (75 µm id × 25 cm length; Thermo Fisher Scientific) at 40 °C. Peptides were separated along a 130 min gradient of 5% to 27.5% mobile phase B [80% (*v*/*v*) acetonitrile (Fisher Scientific, Waltham, USA), 0.08% (*v*/*v*) formic acid (Sigma)] over 105 min followed by a ramp to 40% mobile phase B over 15 min and lastly to 95% mobile phase B over 10 min at a flow rate of 300 nL/min. The mass spectrometer was operated in positive polarity and data-dependent mode (topN, 3 s cycle time) with a dynamic exclusion of 60 s (with 10 ppm error). The RF lens was set at 30%. A full scan resolution using the orbitrap was set at 120,000 and the mass range was set to *m*/*z* 375 to1500. The full scan ion target value was 4.0 × 10^5^, allowing a maximum injection time of 50 ms. The monoisotopic peak determination was used, specifying peptides, and an intensity threshold of 1.0 × 10^4^ was used for the precursor selection. Data-dependent fragmentation was performed using higher-energy collisional dissociation (HCD) at a normalized collision energy of 32 with quadrupole isolation at *m*/*z* 0.7 width. The fragment scan resolution using the orbitrap was set at 30,000, with *m*/*z* 110 as the first mass, ion target value of 2.0 × 10^5^ and a 60 ms maximum injection time.

### 2.3. Protein Search Parameters

Resulting raw files from the analysis of six different *T. truncatus* tissues and raw files from a publicly available 39 fraction HeLa experiment (ProteomeXchange Consortium [26] via the PRIDE partner repository with the dataset identifier PXD004452) were processed and searched using Proteome Discoverer (v.2.0.0.802). For *T. truncatus* analysis, Sequest HT and Mascot (v2.6.0; Matrix Science) search algorithms were used, while only Sequest HT was used for human searches. For all searches, the protein.faa fasta file was retrieved from NCBI RefSeq [27] via ftp (version) [28]. For searches with the prior *T. truncatus* annotation, GCF_000151865.2_Ttru_1.4 was used, while searches with the current *T. truncatus* annotation, GCF_001922835.1_NIST_Tur_tru_v1 was used. These correspond to release 100 and 101 for this organism on NCBI. The whole-genome sequencing projects can be found in GenBank [29] under entries ABRN00000000.2 (Ttru_1.4) and MRVK00000000.1 (NIST_Tur_tru_v1). For the human searches, the following were used: GCF_000001405.10_hg16_Build34.3 (Build 34), GCF_000001405.25_GRCh37.p13 (Release 105; GCA_000001405.14_GRCh37.p13) and GCF_000001405.33_GRCh38.p7 (Release 108; GCA_000001405.22_GRCh38.p7). The *T. truncatus* searches also used the common Repository of Adventitious Proteins database (cRAP; 2012.01.01; the Global Proteome Machine), though these sequences were removed from search results.

The following search parameters were used for Mascot and Sequest: trypsin was specified as the enzyme allowing for two mis-cleavages; carbamidomethyl (C) was fixed and acetylation (protein n-term), deamidated (NQ), pyro-Glu (n-term Q), and oxidation (M) were variable modifications; there was 10 ppm precursor mass tolerance and 0.02 Da fragment ion tolerance. Within Sequest, the peptide length was specified as a minimum of six and maximum of 144 amino acids. The resulting peptide spectral matches were validated using the percolator algorithm, based on q-values at a 1% false discovery rate (FDR). The peptides that were greater than six amino acids long were grouped into proteins according to the law of parsimony and filtered to 1% FDR, and single peptide hits were allowed. Briefly, there may be more than one peptide spectral match for a given peptide, which are then grouped to peptide groups. Protein inference is when these peptide groups are assigned to proteins but given similarity between some proteins (such as isoforms or highly homologous sequences), peptides can match to more than one protein. For this reason, protein families or protein groups are generated based on peptide overlap (and therefore sequence overlap), which reduces inflation due to isoform identifications. For the described analyses, protein and peptide groups are used and are available for each *T. truncatus* search in Appendix A. Raw MS data and Mascot based search results for *T. truncatus*, as well as all fasta databases, have been deposited to the ProteomeXchange Consortium [26] via the PRIDE partner repository with the dataset identifier PXD008808 and 10.6019/PXD008808.

### 2.4. Proteomic-Based Quality Metric for Annotation Quality

Evaluating proteomic results relies on qualifying how well a database explains the observed tandem mass spectra: high numbers of protein identifications and percent identified spectra indicate good proteomic performance. Another way of describing proteomic results is to plot the number of peptide identifications versus the protein molecular weight. A larger protein has potentially more peptide identifications but due to solubilization and digestion effects (such as post-translational modifications and protein folding), larger proteins may not always yield more peptides. For this reason, there is a somewhat Gaussian distribution of peptide frequency around the median protein molecular weight (e.g., Figure 1B). This median can shift right when the molecular weight of predicted protein-coding sequences increases and/or the number of isoforms increases.

When evaluating and comparing de novo genome assemblies and annotations, the specific question that proteomics can answer is the degree of database fragmentation and accuracy. If an annotation improves partial coding sequences to complete protein-coding sequences with isoforms, then there will be an increase in the molecular weight of identified proteins with more peptides assigned to these longer sequences. By simply improving partial sequences, there would be a shift to higher protein molecular weight. One goal of the current study was to provide a more robust quality measure by incorporating unique peptide counts (which correspond to protein coverage) with the change of median molecular weight of inferred proteins. The NP_10_ is a proposed metric that first stratifies the results by identifying the top decile (or 10th 10-quantile) of proteins based on the number of peptides per protein and then returns the median molecular weight of the resulting proteins (graphically demonstrated in Figure 1). To calculate the NP_10_, use a table of identified proteins, number of identified peptides and protein molecular weight, then calculate the top decile based on the number of peptides (Figure 1A) followed by the median molecular weight of this decile (Figure 1B). This approach uses the number of peptides, and not unique peptides; thus, it is possible for peptides to be shared across inferred proteins. Although using unique proteins is important for many quantification approaches, we did not feel it was necessary to apply such strict parsimony. A code is not given, as this is a straightforward process that can be performed in nearly any program. The NP_10_ metric is like simply calculating the median molecular weight of all inferred proteins, but by removing protein identifications with relatively few peptide assignments, it attempts to indicate the accuracy of the improved/longer protein-coding sequences.

### 2.5. Details of Online Resources from NCBI

The National Center for Biotechnology Information (NCBI) has a plethora of excellent data services, and this paper relies heavily on the products from the RefSeq team. As described, the annotations used, for both *T. truncatus* and humans, were created using the NCBI Eukaryotic Genome Annotation Pipeline. Each new annotation has an annotation report, and the report for NCBI *T. truncatus* Annotation Release 101 (which was generated mere days after the NIST Tur_tru v1 assembly was deposited, can be found here, https://www.ncbi.nlm.nih.gov/genome/annotation_euk/Tursiops_truncatus/101/ (accessed on 25 July 2023). Within this report is a section called “Comparison of the current and previous annotations” that contains broad categories of changes used in this paper (Identical, Minor changes, etc.), as well as definitions of these terms, and importantly what these categories correspond to in the hyperlinked tabular report. This report was retrieved and used to map these broad categories onto protein identifications and relate the identifications between the two releases. In addition, NCBI now provides BUSCO scores for genomes, both Genbank assemblies and RefSeq annotations, and this can be found on the newly updated genome pages via the NCBI Datasets portal. There were BUSCO scores available for GCF_001922835.1 (NIST_Tur_tru_v1) but the scores for GCF_000151865.2 (Ttru_1.4) were requested. Details for the BUSCO calculations are as follows: BUSCO v4.0.2 and cetartiodactyla_odb10 (*n* = 13,335). These calculations were performed on the RefSeq releases (GCF), not the Genbank assemblies. Similar BUSCO scores for the now much deprecated human releases are not available.

## 3. Results

### 3.1. Proteomic Analysis of Six Tissues Using NIST_Tur_tru v1

The initial goal of this study was to advance metrological capabilities in *T. truncatus*. This was accomplished by demonstrating proteomic measurements of six tissues from *T. truncatus*. On average, 2199 protein groups and 4888 proteins were identified in each tissue. The reason for performing proteomic analysis on multiple tissue types was to capture more of the possible protein population. Although there were 1310 protein identifications shared across tissues, there was also diversity in protein identifications between tissues, with the brain and skin analyses having the most unique proteins (Figure 2). Proteomic results for each tissue are available (Appendix A). It is interesting to note that the liver, kidney, and blubber came from the individual used for whole-genome sequencing. This dataset is relatively diverse and provides experimental evidence for over 32,000 proteotypic peptides and can be used to generate high-resolution spectral libraries used for the analysis of data-independent acquisition data.

### 3.2. Comparison of Ttru_1.4 and NIST_Tur_tru v1

The second goal of the current study was to evaluate the 2016 *T. truncatus* de novo genome assembly (GCA_001922835.1) and RefSeq annotation (NIST_Tur_tru v1). This genome assembly was generated in the fall of 2016 using shotgun sequencing coupled to an in vitro histone ligation-based sequencing method (i.e., Chicago method) and proprietary assemblers described in detail by Putnam et al. [9]. This process resulted in a genome assembly with a scaffold N50 of 26.6 Mbp (Figure 3A). Of the 584 mammalian species with reference genomes deposited on NCBI as of July 2023, 279 had scaffold N50 values greater than 26.6 Mbp (according to NCBI Datasets). This level of contiguity was becoming more commonplace even back in 2017, with three marine mammal genomes released with a scaffold N50 greater than 19 Mbp (*T. truncatus*; *N. schauinslandi*, Hawaiian monk seal [5]; and *Delphinapterus leucas*, beluga whale [30]). For comparison, the prior NCBI *T. truncatus* RefSeq annotation (Ttru_1.4) was used. This assembly was a 2012 update [23] to the 2008 draft assembly based on Sanger sequencing, Ttru_1.2 [31]. The differences in contiguity, as measured by scaffold N50 and contig N50, are given in Figure 3A. Likewise, BUSCO scores of completeness for each RefSeq annotation are shown in Figure 3B. It is important to note that these BUSCO scores are of the RefSeq annotation and therefore are describing both the assembly and annotation.

The Ttru_1.4 and NIST_Tur_tru v1 genome assemblies are publicly available on NCBI and have been annotated using NCBI’s eukaryotic annotation pipeline and made available in RefSeq [27], as release 100 and 101, respectively. The now deprecated annotation release 101 was based on NIST_Tur_tru v1 and has 24,026 genes and pseudogenes and 17,096 protein-coding genes with 38,849 coding sequences. At the gene and transcript level, there were many changes from Ttru_1.4 that are delineated based on the alignment of genes and transcripts: identical, minor changes, major changes, new, deprecated and other (Figure 4B). These categories are defined and available through NCBI’s annotation report [32]. Briefly, 28% of the prior genes and transcripts in Ttru_1.4 were deprecated, 72% had minor or major changes, and 21% of the genes and transcripts in the NIST release are new. Additionally, a small group of proteins have the prefix YP, which is not included in these NCBI categories.

Tandem mass spectrometry data collected from all six tissues were searched against each annotation release. For both releases, almost 1/3 of the predicted protein-coding sequences were inferred by mass spectrometry. Specifically, the NIST assembly identified 32,582 peptide groups belonging to 10,402 proteins comprising 4711 protein groups. The Ttru_1.4 assembly identified 33,738 peptide groups belonging to 6899 proteins and comprising 5292 protein groups. Many of the differences between the two results were due to a loss of deprecated sequences and minor/major changes (Figure 4B). Broadly, these changes resulted in larger proteins with an increased median molecular weight and NP_10_ molecular weight (Figure 4A).

### 3.3. Confirming Improvements in Gene Annotation

There were 4695 protein-coding sequences in the Ttru_1.4 annotation listed as partial, and one of the main improvements in the 2016 NIST annotation was that 86% of these sequences were merged into complete sequences according to the RefSeq annotation report. This offered an opportunity to evaluate the accuracy of these new assignments by determining whether peptides identified by mass spectrometry supported the new complete sequences. Of 6899 identified proteins using Ttru_1.4, 1249 were partials. Of these, 249 partial proteins identified using Ttru_1.4, 534 had minor changes, 256 had major changes, 450 were deprecated and 9 were other (defined simply as other changes [32]). When this NIST annotation was used, 1005 of these same 1249 proteins were identified, with 985 no longer being listed as partial. The median improvement within each protein was two additional unique peptides; overall, the median molecular weight improved 1.8-fold (Figure 5B,C). Of these 1005 partial proteins identified using Ttru_1.4, when using the NIST annotation, 886 had an increased molecular weight and increased number of unique peptides (Figure 5A).

### 3.4. Comparing Peptide Identifications

An unexpected result in the 2016 annotation was that there were 3% fewer peptide identifications overall, due to the deprecation and updates of sequences (Figure 6). We were interested in tracking these peptide level changes related to the major changes between the two releases related to deprecated genes, new genes, and major changes. Over 80% of the peptide groups identified in NIST annotation were also identified using the Ttru_1.4 annotation (Figure 6). The new peptide identifications were linked to major and minor changes in genes, with only 3.2% due to new sequences. As would be expected, many of the peptide groups not identified in the NIST annotation were deprecated (41%). Given that these 5657 peptide groups lost using the NIST annotation were high-confidence identifications, this may provide evidence for the re-inclusion of these protein-coding sequences in future annotation releases. It is also important to note that the gain and loss of protein sequences are common in reannotations, even though they are compared at a granular level such as this; however, this comparison highlights the potential value in such a comparison.

### 3.5. Specific Examples of Annotation Improvements

The goal of evaluating differences at a broad level is to capture and describe relevant changes at the granular level. At the peptide level, one of the most striking improvements was related to titin, a major component in muscle tissue. In Ttru_1.4, titin (XP_004322250.1) was a partial sequence of 2167 amino acids (241.7 kDa), and 60 unique peptides (40.2% coverage) were identified belonging to this sequence. In the NIST annotation, the coding sequence for titin (XP_019787158.1) was 32,192 amino acids (3812.8 kDa), and 779 unique peptides (34.3% coverage) were identified belonging to this sequence. This single sequence improvement is responsible for many changes observed at the peptide level (Figure 5).

Almost 2% of the identified proteins using the NIST annotation were considered new. One important new protein of note is cystatin C (XP_019783122.1). This protein was not present in Ttru_1.4; meanwhile, when using the NIST annotation, the mass spectrometry data identified three unique peptides (41.3% coverage) belonging to the predicted 13.1 kDa protein. This protein has applications as a biomarker [33], and with these proteomic results, it is possible to create SI traceable mass spectrometer-based assays (similar to [34]). Another protein of note is serotransferrin (XP_019789750.1), which is 90% identical and 3.5% longer than the entry in Ttru_1.4 annotation (XP_004329553.1). Most of these changes were on the c-terminus section (from positions 537 to 634), which was supported by the proteomic data that identified four peptides spanning this region. There were other slight changes to the sequence that resulted in six more unique peptides identified in the improved serotransferrin, which supports the accuracy of the 2016 annotation. Overall, there are many changes related to the over 10,000 protein identifications, many would be considered improvements as indicated by increased protein molecular weight and/or greater peptide coverage. At a gene-by-gene level, these results can be used to confirm and improve annotations.

### 3.6. Confirming Quality Metric in Human Annotations

To gauge the broader applicability of using proteomics as a quality measure of genomic annotations, we demonstrated NP_10_ in a more mature genome with deeper proteomics. The work by Bekker–Jensen et al. [20] is publicly available on ProteomeXchange [26,35]; for this comparison, the data generated from a 39 fraction high pH pre-fractionation of a HeLa cell digest followed by an LC-MS/MS analysis was used for database searching. These data were searched against three human genome annotations from 2004, 2013 and 2016, each with markedly increased scaffold N50 values and database sizes (i.e., number of coding-sequences; Table 1). Unfortunately, BUSCO completeness scores were not available for these releases. The HeLa dataset was searched against each RefSeq annotation. The number of identified proteins was 13,341, 22,906 and 48,019 proteins in Build 34, Release 105 and Release 108, respectively. The median molecular weight of identified proteins increased 25% (from 51.06 to 53.46 to 63.99 kDa, respectively) whereas the increase in NP_10_ was more pronounced with a 33% improvement (from 100.17 to 101.87 to 133.55 kDa, respectively; Figure 7).

## 4. Discussion

Advances in bioanalytical platforms across domains (i.e., genomics, transcriptomics, and proteomics) are improving the accessibility of non-model organisms as viable research candidates. The results of the current study provide secondary confirmation of 10,402 proteins from 4711 protein groups using a recently completed well-scaffolded high-coverage *T. truncatus* genome and bottom-up proteomic analysis of six different tissues. Previous proteomic studies of *T. truncatus* have identified less than 100 protein groups in serum [24,34], while the most detailed published proteomic analysis of a marine mammal identified 206 proteins in the cerebrospinal fluid of *Zalophus californianus* (California sea lion) [36]. As of July 2023, according to NCBI Datasets, there were 12 cetacean and 10 pinniped genomes that have been annotated by NCBI RefSeq (of the 584 mammalian species with reference genomes currently deposited on NCBI), though only *Mirounga angustirostris*, *Phoca largha*, *T. truncatus* and *Z. californianus* have published mass spectrometry-based proteomic datasets on ProteomeXchange. Work is underway to increase the number of marine mammal genomes along with companion high-quality proteomic datasets. The results of the current study provide empirical confirmation of protein annotations, including observable proteotypic peptides, which can be a resource for future targeted studies in *T. truncatus*. For example, by improving the protein-coding sequence accuracy of serotransferrin in *T. truncatus,* future studies can extrapolate metrological advances in human serotransferrin sialoforms [37] to *T. truncatus* disease treatment [38]. Since the current results are not an exhaustive proteomic dataset, future studies will utilize different solubilization techniques, proteases, and separation techniques to provide even deeper proteome coverage (reviewed and demonstrated in the following [20,39,40]). Still, it is worth noting that in single study using a simple experimental approach, we have identified almost 1/3 of the possible predicted proteins, emphasizing the ease of accomplishing bioanalytical advances in non-model organisms using modern techniques.

In the current study, benchmark proteomic datasets were used to evaluate genome assembly and annotation improvements in *T. truncatus* and humans. Typically, a reference database is used to demonstrate proteomic improvements due to optimized protein extraction, solubilization and digestion, peptide separation, mass spectrometer speed and mass accuracy, search algorithm performance and database accuracy. In contrast, when the mass spectrometric data are held constant and the database is varied instead, differences in proteomic results are indicative of database fragmentation and accuracy. A proteomic analysis of multiple tissues allowed for greater protein diversity when evaluating *T. truncatus*, though the publicly available human data performed exceptionally well despite using a single tissue, since it utilized highly optimized separation techniques. The benefit of using multiple tissues was a diverse sampling of the proteome (for example, titin is highly abundant in muscle tissue), but the high degree of fractionation in the human study resulted in more proteins overall (12,209 proteins) [20]. An optimum proteomic benchmark dataset would be one that offers the possibility of the deepest proteome coverage. This would rely on using multiple tissues, extraction protocols, enzymes and optimum separation techniques coupled with modern mass spectrometers, such as the recent study by Sinitcyn et al. [21]. These benchmark species-specific proteomic datasets could be developed in parallel to the exponential increase in de novo genomes being released and annotated and would prove invaluable in exercises assessing assembly and annotation performance (such as Assemblathon 2 [8]). Importantly, given the abundance and accessibility of public proteomic data in this “Golden Age of Proteomics” (as coined by [41]) and modular open-access proteogenomic pipelines such as Galaxy-P [42,43], it would be possible to incorporate these reference mass spectrometric datasets and proteomic-derived quality metrics into genome assembly and annotation pipelines.

In parallel to improvements in genome assembly contiguity and annotation accuracy, proteomic results should have increased peptide numbers per protein, higher protein identifications due to isoform resolution and improved coverage of higher molecular weight proteins due to better long-range accuracy. For instance, when evaluating the substantial reduction in partial sequences between Ttru_1.4 and NIST_Tur_tru v1, there was an increase of 81% in the median molecular weight of these proteins that coincided with more peptide identifications within these new complete sequences. The most drastic example in this case study was titin, which went from 60 to 779 identified peptides with the addition of over 32,000 amino acids to the previously partial sequence. This also emphasizes that greater numbers of protein identifications do not imply a higher quality, since a more fragmented genome may provide more protein identifications. Instead, the identification of larger proteins with more identified peptides is more indicative of an improved quality. This concept is akin to BUSCO scores [10,13,14,15], which summarize the length of ortholog genes present in a genome assembly (i.e., complete, duplicated, fragmented, or missing). Instead of using a database (OrthoDB [16]) of orthologous sequences determined for a given clade to analyze a genome assembly, we are utilizing empirical data of peptide fragmentation to evaluate the genome annotation. In other words, instead of asking how well a genome aligns with known orthologs (i.e., completeness), we are asking how well a genome explains proteomic data of the species itself (i.e., accuracy). The proposed metric, NP_10_, attempts to capture this quality measure beyond numbers of peptide spectral matches or protein identifications. In this case study of *T. truncatus*, we quantify genome assembly improvements in contiguity (200-fold improvement in scaffold N50), genome assembly and annotation improvements in completeness (6% improvement in BUSCO complete score) and genome annotation improvements in accuracy (21% improvement in NP10). Still, these measures gloss over the granular changes that later affect specific protein identifications; however, we think the change in NP_10_ is reflective of genome accuracy, which is essential to the mass spec-based proteomic analysis. Lastly, as we showed in the *T. truncatus* example, there are sequences that may be lost in newer annotations that could in fact be valid based on peptide spectral matches (Section 3.4). There is an opportunity to develop a streamlined method to track MS/MS spectra assignments and quantify those changes with database improvements to establish finer measures of search space effects on the proteomic performance.

## 5. Conclusions

With constant improvements in genome assemblies and annotations, the databases underpinning mass spectrometry-based proteomic analysis are constantly evolving, yet the gains and changes are rarely quantified or investigated. Though it may be interesting to evaluate each human release at the level of detail we have provided for *T. truncatus*, the NP_10_ metric provides a coarse measure of improvements. Overall, these results demonstrate that modern whole-genome sequencing techniques can provide high quality de novo genome assemblies and that proteomics are a useful metrological tool to evaluate genome annotation and benchmark genome accuracy, which can complement existing measures of genome completeness, such as BUSCO.

## Figures and Tables

**Figure 1 genes-14-01696-f001:**
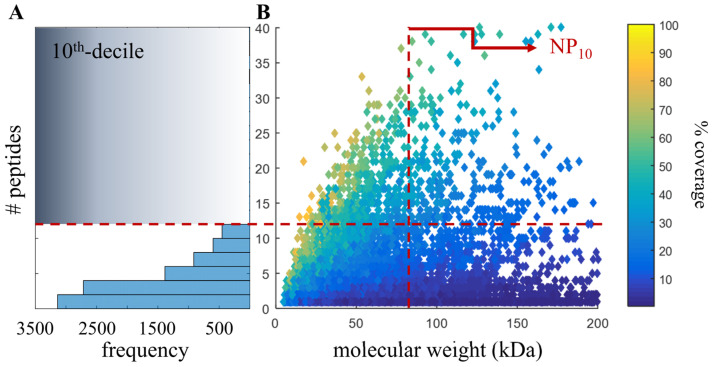
Graphical example of NP_10_ calculation. Bottom-up proteomics uses tandem mass spectrometer data to identify peptides that are then used to infer proteins, both of which rely on a given database. For every protein identified, there is a number of peptides identified that belong to that protein; the amount of protein sequence these peptides span (percent coverage), and the calculated molecular weight of the identified protein (in kilodaltons; kDa), is shown. The NP_10_ is a proposed metric that first stratifies the results by (**A**) identifying the top decile (or 10th 10-quantile) of unique peptides per protein. In this example, 12 peptides, shown by the red dotted line. (**B**) Next, this number of peptides is used to return those proteins with at least the top decile (or 10th 10-quantile) of peptides per protein. Finally, from this protein subset, the median molecular weight is calculated (in this example, 82 kDa), shown by the red dotted line. Although percent coverage is shown, it is not used in this calculation.

**Figure 2 genes-14-01696-f002:**
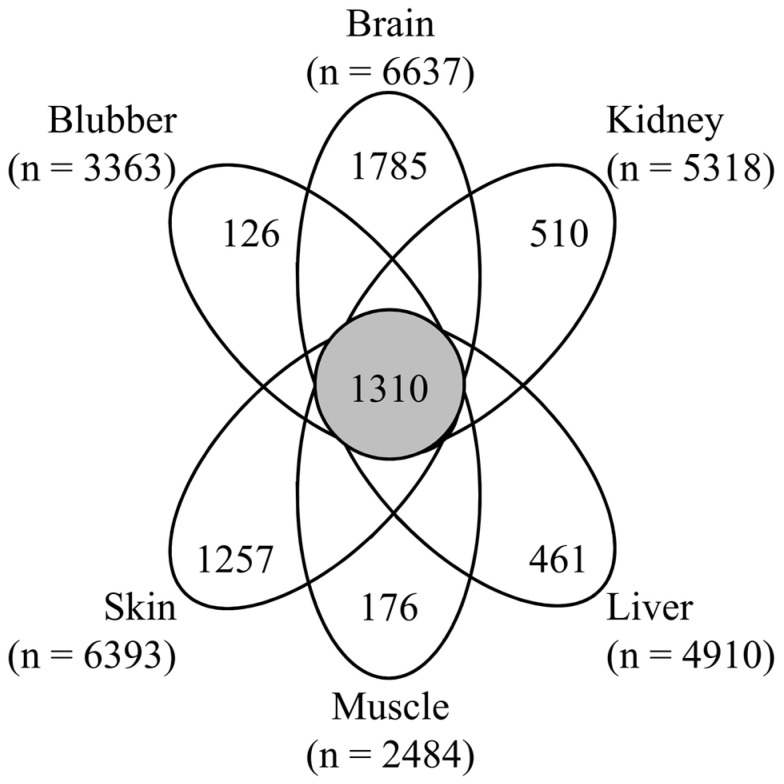
Overlap and unique protein identifications by *T. truncatus* tissue. Proteins unique to each tissue and shared by all tissues are shown along with the total number of proteins identified in each analysis.

**Figure 3 genes-14-01696-f003:**
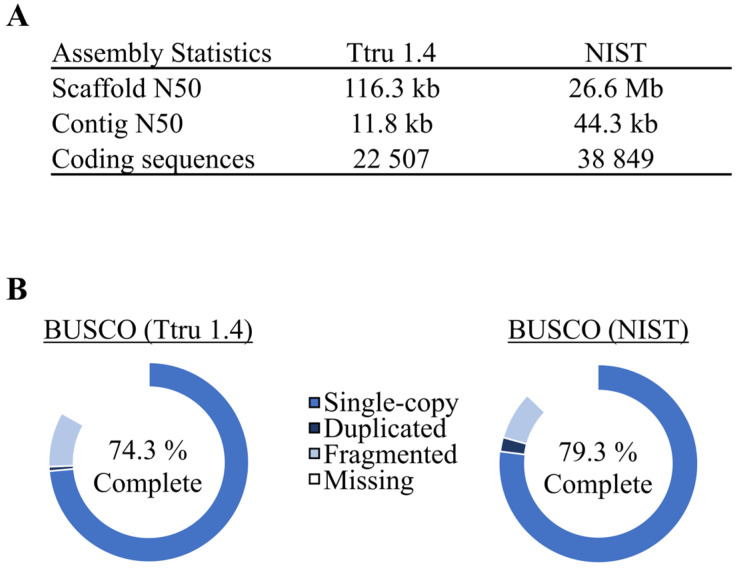
Descriptive statistics and BUSCO scores of both *T. truncatus* genomes. (**A**) Genome assembly statistics related to contiguity, as well as the total number of coding sequences in the annotation. (**B**) BUSCO completeness scores of both RefSeq annotations against the cetartiodactyla_odb10 dataset (*n* = 13,335). The complete score is the sum of the single-copy and duplicated values. The complete BUSCO scores for Ttru_1.4 are: single-copy—73.6%, duplicated—0.7%, fragmented—8.9%, and missing—16.8%. The BUSCO scores for NIST Tur_tru v1 are: single-copy—77.0%, duplicated—2.3%, fragmented—7.9%, and missing—12.8%. Overall, there was a 6% increase in BUSCO completeness.

**Figure 4 genes-14-01696-f004:**
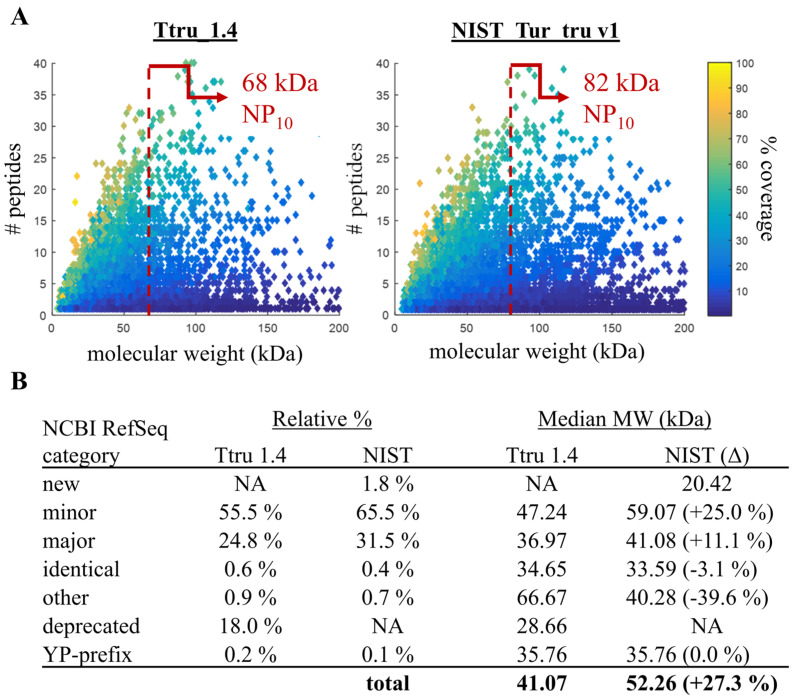
Descriptive statistics of identified proteins using different annotations. (**A**) The NP_10_ molecular weight improved 21.3% from 67.59 kDa to 81.99 kDa (indicated by the red dotted line) with the updated genome assembly and annotation. (**B**) There was also an improvement in median molecular weight (MW) of inferred proteins across genes with minor and major changes. (Note: these axes have been truncated for illustration and do not show all data points.).

**Figure 5 genes-14-01696-f005:**
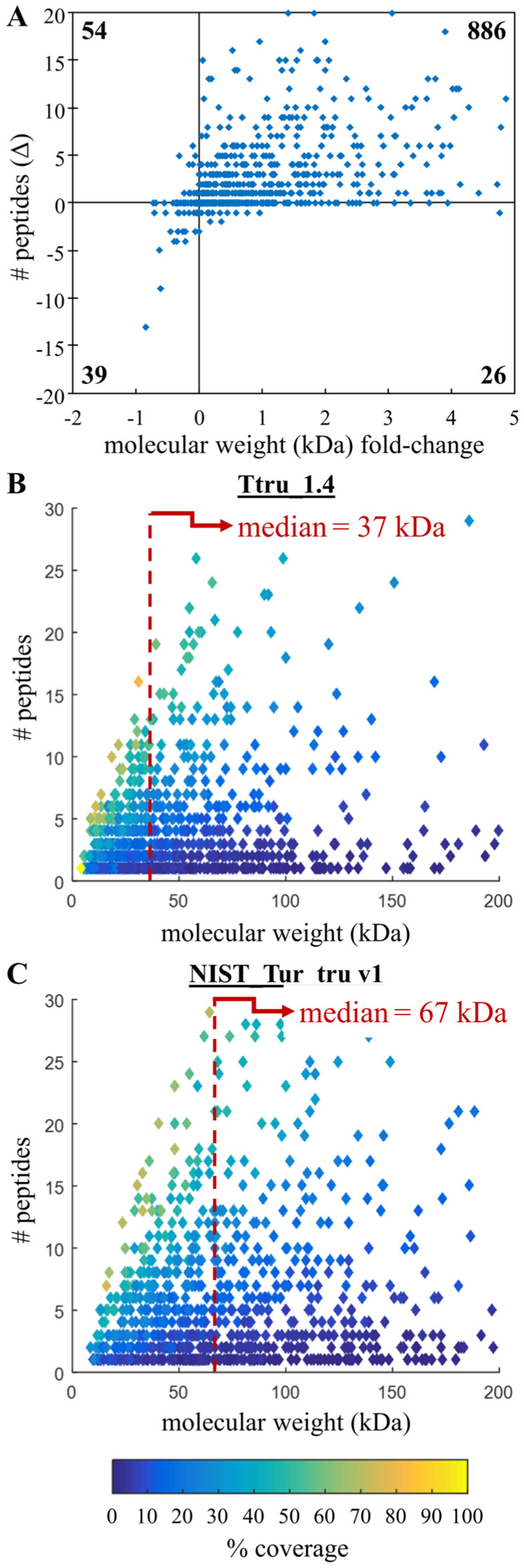
Confirming improved annotation of former partial proteins. (**A**) Proteins that were partial in the Ttru_1.4 annotation were improved in the NIST annotation with increased molecular weight and number of identified peptides with the sum of each quadrant shown in bold, and (**B**,**C**) there was mass spectrometric evidence to support the accuracy of these improvements corresponding to increased peptide identifications and median molecular weight (the latter indicated by the red dotted line; note: these axes have been truncated for illustration and do not show all data points).

**Figure 6 genes-14-01696-f006:**
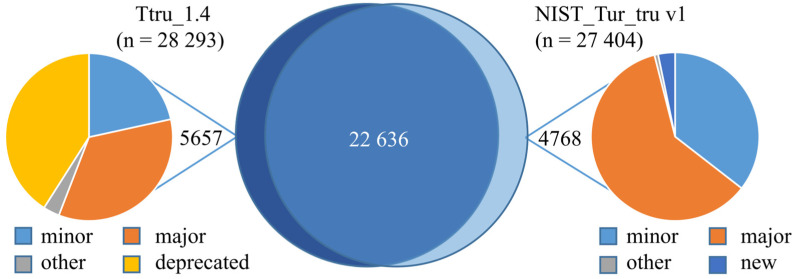
Source of peptide identification differences using the two assemblies. There was strong overlap of identified peptides using the two assemblies with over 80% overlap. The sources of the differences were largely comprised of deprecated proteins in Ttru_1.4 (41% of the 5657) and minor/major changes in NIST_Tur_tru_v1 (96% of the 4768).

**Figure 7 genes-14-01696-f007:**
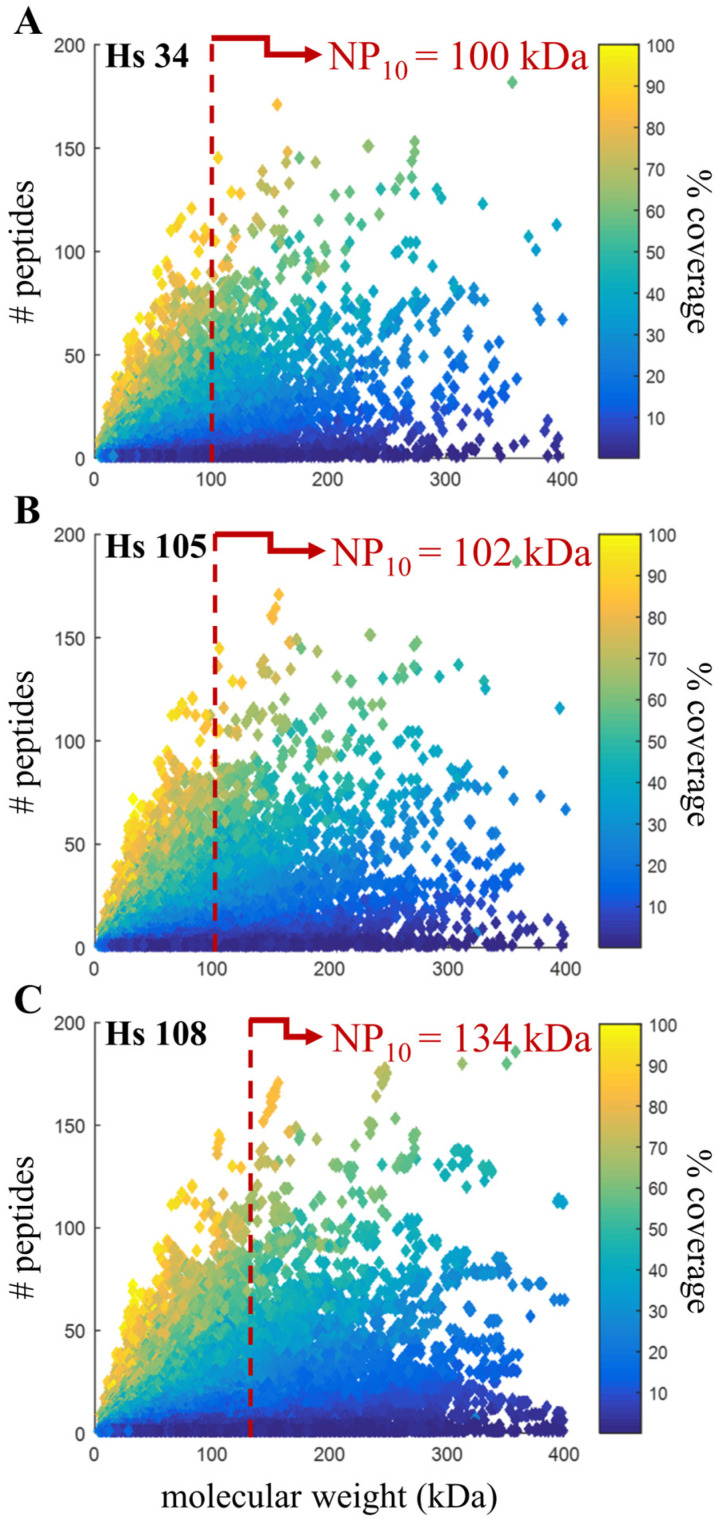
Similar trends with improved human assemblies. As the contiguity of the human genome has improved from (**A**) Build 34 (in 2004) to (**B**) Release 105 (in 2013) to (**C**) Release 108 (in 2016), there is a shift upward and to the right, indicating that annotations are more accurate (increased coverage) and complete (increased molecular weight). The NP_10_ improved by 33% and is indicated by the red dotted line (note: these axes have been truncated for illustration and do not show all data points).

**Table 1 genes-14-01696-t001:** Descriptive statistics of human-annotated databases and resulting proteomic identifications. Hs 34, Hs 105 and Hs 108 refer to human (*Homo sapiens*) Build 34, Release 105 and Release 108, respectively.

	Hs 34	Hs 105	Hs 108
Release date	Feb 2004	Jun 2013	Mar 2016
Assembly name	NCBI 24	GRCh37.p13	GRCh38.p7
Scaffold N50	29.1 Mbp	45.0 Mbp	59.4 Mbp
Coding sequences	27,180	45,107	109,018
Protein groups identified	9762	10,059	10,219
Proteins identified	13,341	22,906	48,019
Peptide groups identified	175,895	184,580	184,806
Peptide spectral matches	390,909	405,852	405,950
NP_10_	100.17 kDa	101.87 kDa	133.55 kDa

## Data Availability

The raw data and tissue specific search results along with all databases used are available at the ProteomeXchange Consortium [26] via the PRIDE partner repository with the dataset identifier PXD008808 and 10.6019/PXD008808. The proteomic data from Bekker–Jensen et al. [20] used for the human comparison can be found at ProteomeXchange Consortium [26] via the PRIDE [44] partner repository with the dataset identifier PXD004452. Tabulated search results for combined analysis and for each tissue can be found in Appendix A.

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
