# Peer review of "Proteomics as a Metrological Tool to Evaluate Genome Annotation Accuracy Following De Novo Genome Assembly: A Case Study Using the Atlantic Bottlenose Dolphin (Tursiops truncatus)"

_genes, 2023, doi:10.3390/genes14091696_

Round 1

Reviewer 1 Report

In this MS, Neely et al. have unveiled a compelling approach to assess genome annotation accuracy using proteomics in their study of the Atlantic bottlenose dolphin. The introduction of the NP10 metric is innovative and reflects an interesting aspect of quality measurement. The results are promising, with a 21% improvement in NP10 for the T. truncatus genome, signifying proteomics as a potent tool in this field. The design is methodologically robust and the manuscript offers valuable insights into both existing and potential future applications of proteomics in non-model organisms. Though the paper is well-written and substantiated with relevant data, I have some minor concerns and suggestions that could further refine and clarify certain sections.

Line 11. This sentence is somewhat complex. Consider restructuring for clarity, such as: "Using the 2016 Atlantic bottlenose dolphin NCBI RefSeq annotation and mass spectrometry-based proteomic analysis of six tissues, we confirmed 10,402 proteins from 4,711 protein groups, constituting nearly one-third of the possible predicted proteins."

Line 29. It might be beneficial to include some quantitative information or references that detail the scale of cost reduction and improvement in read lengths.

Line 88. It could be strengthened by briefly mentioning why this specific organism was chosen.

Line 201. "… larger proteins do not always yield more unique peptides. (REF)"

Line 239. Revise to "2,199 protein groups and 4.888 proteins"

Line 315. The authors observed a decrease in peptide identifications in the 2016 annotation, mentioning major changes in genes. However, a more detailed explanation of how these changes led to the decrease would provide clarity.

Line 410. This section discusses the advantages of proteomic analysis across multiple tissues but seems to lack a detailed exploration of why human data performed exceptionally well despite the utilization of a single tissue.

Line 435. The concept of NP10 is introduced as a quality metric, but the text lacks a comprehensive explanation of how it overcomes the limitations of existing metrics, and there is no in-depth discussion of potential differences between it and other metrics.

Line 438. A streamlined method for tracking MS/MS spectra assignments and quantifying those changes with database improvements has been mentioned, which appears promising, but is somewhat vague.

Reviewer 2 Report

The authors present a novel approach to evaluate genome annotation accuracy using proteomics, and benchmark it on the Atlantic bottlenose dolphin (Tursiops truncatus). The findings suggest that proteomics may serve as a useful tool to verify genome accuracy, emphasizing the need for reference proteomic datasets across species.

The study is well-performed and relevant. I recommend it for publication, provided that the following minor concerns are addressed:

- In the introduction, lines 60-65, the authors introduce other methods to evaluate genome completeness. Can the authors expand on how their method compares to existing methods in the Discussion section of the manuscript? For example, what are the advantages and disadvantages of using NP10 compared to BUSCO?

- In Figure 1 it is unclear whether the notation "# peptides" refers to total number or number of unique peptides. Can the author clarify, in the figure, legend, and Results text.

- Protein coverage by MS is essential to the discussion of the NP10 techniques. The manuscript could benefit from further references on the topic. Specifically, in the Introduction, line 82, I recommend the authors to mention the importance of peptide pre-fractionation techniques to obtain more peptides and increase protein sequence coverage, and include the following reference:

Panizza et al., "Isoelectric point-based fractionation by HiRIEF coupled to LC-MS allows for in-depth quantitative analysis of the phosphoproteome", Sci Rep, 2017

- Figures 1, 3, 5, 6: please add letter notations for figure panels (A, B, C, etc). Break down the figure legends to include panel-by-panel descriptions. Also refer to specific figure panels in the main text (Results and Discussion).
